# Pyrrolizidine Alkaloids in Food on the Italian Market

**DOI:** 10.3390/molecules28145346

**Published:** 2023-07-11

**Authors:** Mariantonietta Peloso, Gaetan Minkoumba Sonfack, Sandra Paduano, Michele De Martino, Barbara De Santis, Elisabetta Caprai

**Affiliations:** 1National Reference Laboratory for Plant Toxins in Food, Food Chemical Department, Istituto Zooprofilattico Sperimentale della Lombardia e dell’Emilia Romagna, Via Fiorini 5, 40127 Bologna, Italy; m.peloso@izsler.it (M.P.); g.minkoumbasonfack@izsler.it (G.M.S.); 2Ministry of Health, General Directorate for Hygiene and Food Safety and Nutrition, Via G. Ribotta, 5, 00144 Rome, Italy; s.paduano@sanita.it (S.P.); m.demartino@sanita.it (M.D.M.); 3National Reference Laboratory for Plant Toxins in Food, Food Chemical Department, Istituto Superiore di Sanità, Viale Regina Elena, 299, 00161 Rome, Italy; barbara.desantis@iss.it

**Keywords:** pyrrolizidine alkaloids, fresh borage leaves, dried tea, dried herbs, dried herbal infusions, pollen, honey, lycopsamine-type

## Abstract

Pyrrolizidine alkaloids (PAs) are secondary metabolites produced by over 6000 plant species worldwide. PAs enter the food chain through accidental co-harvesting of PA-containing weeds and through soil transfer from the living plant to surrounding acceptor plants. In animal studies, 1,2-unsaturated PAs have proven to be genotoxic carcinogens. According to the scientific opinion expressed by the 2017 EFSA, the foods with the highest levels of PA contamination were honey, tea, herbal infusions, and food supplements. Following the EFSA’s recommendations, data on the presence of PAs in relevant food were monitored and collected. On 1 July 2022, the Commission Regulation (EU) 2020/2040 came into force, repealed by Commission Regulation (EU) 2023/915, setting maximum levels for the sum of pyrrolizidine alkaloids in certain food. A total of 602 food samples were collected from the Italian market between 2019 and 2022 and were classified as honey, pollen, dried tea, dried herbal infusions, dried herbs, and fresh borage leaves. The food samples were analyzed for their PA content via an in-house LC-MS/MS method that can detect PAs according to Regulation 2023/915. Overall, 42% of the analyzed samples were PA-contaminated, 14% exceeded the EU limits, and the items most frequently contaminated included dried herbs and tea. In conclusion, the number of food items containing considerable amounts of PAs may cause concern because they may contribute to human exposure, especially considering vulnerable populations—most importantly, children and pregnant women.

## 1. Introduction

Pyrrolizidine alkaloids (PAs) are a group of natural toxins exclusively biosynthesized as secondary metabolites by over 6000 plant species worldwide [1], created as a defense against insect herbivores due to their inhibitory effect on acetylcholinesterase [2,3,4,5]. PAs are regarded as undesirable substances in food and feed [6]. The European Food Safety Authority (EFSA) CONTAM Panel has identified the following PAs to be of particular importance for food and feed: senecionine-type (in Asteraceae and Fabaceae family), lycopsamine-type (in Asteraceae and Boraginaceae family), heliotrine-type (in Boraginaceae family), and monocrotaline-type (in Fabaceae family) (Table 1) [1]. PAs occur in different parts of the plants, especially in the seeds and inflorescences, and their content depends on many factors (species, plant organ, harvest, and climatic conditions) [1,7]. The contamination pathways of plant-derived products include the accidental co-harvesting of PAs-containing weeds and natural horizontal transfer through the soil [8,9,10,11]. The central structure of PA consists of a necine base (two-fused five-membered rings joined by a single nitrogen atom) and one or more necic acids (mono- or dicarboxylic aliphatic acids) [1,3]. In addition, PAs frequently co-occur in two forms, tertiary amine and corresponding N-oxide (PANOs) [1]. In animal studies, 1,2-unsaturated PAs have proven to be hepatotoxic, pneumotoxic, genotoxic, and carcinogenic substances and exhibit developmental toxicity [1,10,12,13]. These molecules are promptly absorbed in the gastrointestinal [13] tract and metabolically activated in the liver to exert their toxicity, which is the target organ. PA intake is considered to be one of the major causes of hepatic veno-occlusive disease (HVOD) [10,14,15]. There are three activation pathways: hydrolysis and N-oxidation, which promote the excretion of PAs, and oxidation by cytochrome P-450 monooxygenase, which is responsible for their high toxicity as it leads to the transformation of these compounds into reactive pyrroles, which can react with proteins and form DNA adducts [1,3,10,12]. The International Agency for Research on Cancer (IARC) has classified lasiocarpine, monocrotaline, and riddelliine as being possibly carcinogenic to humans—group 2B—whereas other molecules were included in Group 3 as not classifiable as carcinogenic to humans due to limited evidence in experimental animals [13,16]. In recent years (2020–2023), PA alerts collected in the Rapid Alert System for Food and Feed (RASFF) EU portal have significantly increased, especially in food such as oregano, cumin, tea, and pollen [17]. Since 2007, the EFSA has published five scientific opinions about the risk to human health related to the presence of PAs in different types of food [1,6,14,16,18]. According to the 2017 EFSA statement, the most contaminated foods were honey, tea, herbal infusions, and food supplements. Even if the actual exposure levels of the population to these natural contaminants are still uncertain due to a lack of data, the EFSA recommended continuing to collect occurrence data and monitoring a set of 17 PAs and their N-oxides identified as relevant [16]. On the 1st of July 2022, the EU Commission Regulation 2020/2040 was put into force, repealed by the EU Commission Regulation 2023/915 (25 April 2023) [19]. It establishes new maximum levels of 35 PAs in a selection of food items. These 35 PAs (Table 1) extends the list of PAs recommended by EFSA to 21 molecules and considers 14 co-eluting isomers [20]. Following the EFSA’s recommendations and Commission Regulation (EU) 2020/2040 [16,20], analytical data on the presence of PAs have been monitored in 602 food samples collected from the Italian market between 2019 and 2022. Analyses were performed using an in-house LC-MS/MS method, developed and validated by the National Reference Laboratory for Plant Toxins in Food (LNR-TVN) at the Food Chemical Department of the Istituto Zooprofilattico Sperimentale della Lombardia e dell’Emilia-Romagna (IZSLER) in Bologna.

## 2. Results and Discussion

### 2.1. LC-MS/MS Method Validation

The LC-MS/MS method was validated for pyrrolizidine alkaloids and applied to honey, bee pollen, dried tea, dried herbal infusions, dried herbs, fresh borage leaves, and to other matrices not included in this work (cumin seeds and liquid products such as tea and herbal infusions). The Commission Regulation (EU) 2023/915 sets maximum levels for the sum of 21 PAs and the following 14 additional PAs isomers that may co-elute with one or more of the 21 PAs: indicine/rinderine/echinatine, indicine-N-oxide/rinderine-N-oxide/echinatine-N-oxide, integerrimine, integerrimine-N-oxide, heliosupine, heliosupine-N-oxide, spartoidine, spartoidine-N-oxide, usaramine, and usaramine-N-oxide [19]. This method allowed us to detect and quantify the 21 PAs of the Regulation and 4 additional isomers. The coelution was tested for all the isomers included in the Regulation and was confirmed for 10 PAs isomers, while for the following 4 isomers, good chromatographic separation was achieved: integerrimine-N-oxide, echinatine-N-oxide, rinderine-N-oxide, and heliosupine-N-oxide. In the chromatograms of each matrix blank extract, no interfering peaks were detected at the retention times of all PAs.

For validation, the analyzed samples were classified into three categories: honey (multi-floral, *Robinia pseudoacacia*, *Castanea sativa*, and *Heliantus annuus* L.), bee pollen (*Hedera helix*, *Castanea sativa*, and multi-floral), and dried tea–dried herbal infusions–dried herbs–fresh borage leaves (*Camellia sinensis*—green and black tea, *Matricaria chamomilla* L., *Thymus vulgaris* L., and *Borago officinalis* fresh leaves).

Good linearity was observed for concentrations of each analyte between 1 and 50 µg/kg in honey, 5 and 1000 µg/kg in pollen, and 5 and 2000 µg/kg in dried tea, dried herbal infusions, dried herbs, and fresh borage leaves; the linear regression coefficient (R^2^) was ≥0.99 for all analytes in all matrices. According to the EURL-MP guidance document for plant toxins performance criteria, a recovery range of 70–120%, a repeatability relative standard deviation (RSDr) < 20%, and an RSD_R_ of within-laboratory reproducibility < 20% were considered acceptable for every individual toxin [23]. Therefore, the method can be considered fit for purpose. The limits of quantification (LOQs) for all analytes were 1 µg/kg for honey and 5 µg/kg for all other matrices. The validation parameters are summarized in Appendix A.

### 2.2. Occurrence of PAs/PANOs in Food

The LC-MS/MS method was applied to analyze 602 food samples from all the Italian regions between 2019 and 2022 in the following categories: honey, bee pollen, dried tea, dried herbal infusions, dried herbs, and fresh borage leaves. Samples were analyzed for their PA content, and 21 target analytes and 4 additional isomers were investigated. The quality control (QC) sample for each matrix was assessed simultaneously with 70–120% recovery. Of all the food samples analyzed, 42% were ≥LOQ, of which 14% exceeded the maximum levels set by EU Regulation 2023/915 [19]. The analysis showed that the trend of the percentages of contaminated samples has increased from 2019 to the present day, in line with EU RASFF notifications, particularly for oregano and cumin seeds [17].

The PA concentration ranges and the predominant analyte found per matrix are shown in Table 2, and the chromatograms of the positive samples with the predominant analytes and the corresponding spiked blank samples for each matrix are shown in Appendix A.

Dried tea and dried herbs were found to be the most contaminated, in addition to borage, a PA-producing plant. Among these, dried herbs showed the highest percentage of samples that exceeded the EU proposed maximum limit, with 21% of the contaminated samples exceeding the maximum level (1000 µg/kg) (Figure 1). In the majority of the food items investigated, the predominant PAs present in the samples were classified as lyc-type (Table 1 and Table 2). Dried tea and herbal infusions were characterized by the presence of a high amount of sen-type PAs. The PAs belonging to the lyc- and sen-type are typical of the PA-producing plants widespread in the Mediterranean area (*Echium* spp., *Senecio* spp., *Borago officinalis*, and *Eupatorium* spp.) [24,25,26].

As shown in Table 2, the PANOs were the most predominant alkaloids in the samples. Hartmann et al. [5] suggested that PANOs are easily reduced to the pro-toxic free bases in the digestive system of bees. For this reason, PAs generally occur as free bases in honey and as N-oxides in plants [27]. In addition to honey, PAs may also contaminate other animal-derived food. PA-containing plants may be present in products used as animal feed, and several studies have shown that PAs occur in meat, eggs, and milk, although in low amounts [1,10,28,29]. The reason for the contamination of plant families where PAs are unpredictable is accidental harvesting of PA-containing weeds and natural horizontal transfer between plants through soil [8,9], so cereals and other plants not included in Regulation EU 2023/915 should also be monitored.

#### 2.2.1. Honey Samples

Out of 320 honey samples analyzed—of which 21 were monofloral and 299 were multi-floral—94 were found to be contaminated. No maximum level has been set for honey by the EU Commission [19]. The highest value found was 121.1 µg/kg, the mean was 15 µg/kg, and the lowest value was 1 µg/kg, corresponding to the LOQ. According to other research on Italian honey [30,31,32], echimidine was the predominant analyte found in this study. Among the monofloral honey samples, the highest PA value found was 11 µg/kg (Appendix A) in a *Stachys officinalis* honey sample, which was exclusively due to echimidine. In the multi-floral honey, the highest PA content was 121.1 µg/kg, and the highest amount of echimidine was 45.3 µg/kg. In a previous study [33] of IZSLER in Bologna, 121 honey samples were analyzed, 31% of which were contaminated with PAs in the range of 0.9–33.1 µg/kg. The honey EU and extra-EU samples analyzed by Martinello et al. were also contaminated with a greater presence of echimidine (the highest value was 169 µg/kg) [30].

Dübecke et al. observed differences in the amount of PAs in honey depending on the origin country [34]. Kast et al.’s work is one of the few studies that have investigated the botanical origin of honey—in this case, specifically focusing on the botanical origin of Swiss honey. They detected low concentrations of PAs in honey and observed that the most common alkaloids were typical of the genus *Echium* spp.—such as echimidine—widespread in Switzerland and in many regions of northern Italy [25]. In the study by Picron et al. [24], honey samples from Belgium and Mediterranean countries such as Spain, France, Greece, and Turkey were analyzed, and in line with the results of the other studies, a predominance of lycopsamine- and heliotrine-type PAs was observed.

The pollen produced by bees from PA-producing plants is the most common accidental source of PA contamination in honey samples [10,33,35,36]. However, pollen may also be accidentally introduced into the honey by beekeepers during the production process [10,37]; for instance, beekeepers in many countries regularly use some PA-producing plants for honey production, as reported by Moreira et al. [3].

#### 2.2.2. Bee Pollen Samples

Of 130 bee pollen samples analyzed, 66 had a PA-content ≥ LOQ, and 13 exceeded the established maximum level (500 µg/kg), with values ranging from 501 µg/kg to 10,168 µg/kg [20]. The mean total amount was 773 µg/kg, and the lowest was 6 µg/kg, above the LOQ. The predominant analyte was echinatine-N-oxide, a lyc-type analyte typically found in species of the Boraginaceae and Asteraceae families, as confirmed in the work of Fedrizzi et al., who studied Italian pollen samples from the Emilia-Romagna region over three consecutive years (2020–2022) in different seasons. Pollen analysis showed that the source of PAs was probably due to small percentages of flower pollen grains derived from PA-producing plants, such as *Echium* spp. and *Eupatorium cannabinum* from the Boraginaceae family [38]. Of the PA-producing plants, the most present in Italy are *Echium vulgare*, *Echium plantagineum*, *Echium italicum*, *Eupatorium cannabinum*, and *Senecio* spp. [26,38]. There are several studies about PA content in commercial pollen samples in several European countries. The highest PA levels have been found in bee pollen from Mediterranean countries, where PA-producing plants are more widespread. Kempf et al. [35] reported an average PA content of 5170 µg/kg, and Dübecke et al. [34] reported that of 1846 µg/kg, with a predominance of PAs typical for *Echium* spp., while in the study by Kast et al. [39], 31% of the Swiss pollen samples analyzed had a mean PA content of 319 µg/kg, mainly *Echium* spp.- and *Eupatorium* spp.-type PAs, and the remaining 69% of the pollen samples had a PA amount below 1–3 µg/kg. According to the authors, compared to the Mediterranean area, where many species of *Echium* are widespread, these lower levels may be due to the presence of only *Echium vulgare* in Switzerland. The study by Martinello et al. analyzing pollen from northern Italy, whose flora is more similar to that of Switzerland, also confirms these lower values; they discovered an average PA content of 97 µg/kg, and the highest value detected was 271 µg/kg [40].

#### 2.2.3. Dried Tea Samples

Dried tea (*Camellia sinensis*) is the most representative of the samples with the highest levels of contamination. Of the 74 samples analyzed, 44 showed the presence of PAs. Eleven samples exceeded the maximum level of the EU Regulation (150 µg/kg), with values between 167 and 1346 µg/kg [20]. The mean PA content was 158 µg/kg; the lowest was equal to the LOQ, 5 µg/kg. The predominant analyte was retrorsine-N-oxide. The chromatograms of retrorsine-N-oxide (374.2 µg/kg) in a positive dried tea sample are shown in Appendix A. The results of this investigation are in line with other studies [40,41], e.g., the scientific report by Mulder et al. showed that PAs in the tea samples were classified as sen-type, and retrorsine- and senecionine-N-oxide were the most prevalent alkaloids, detected in 60% of all samples [6]. A very high level (4246 µg/kg) was found in green tea by Picron et al. [42], and in this study, most of the samples were PA-contaminated (88–91%) with retrorsine-N-oxide. The predominance of these PAs suggests that the main source of contamination may be accidental co-harvesting of *Senecio* spp. [42]. The Huybrechts and Callebaut study [43] was one of the first to confirm tea as a major source of PAs in samples from the EU market. In contrast to our study, the Belgian market samples showed a predominance of lyc-type PAs due to possible contamination by plants of the Boraginaceae and Asteraceae families.

#### 2.2.4. Dried Herbal Infusion Samples

Of 41 dried herbal infusions samples analyzed, 21 had a PA content ≥ LOQ. Six samples exceeded the maximum level (200 µg/kg) [20] with values between 216 and 1171 µg/kg. Three samples were chamomile, with values between 216 and 268 µg/kg; one was lemon balm, 218 µg/kg, and two were other herbal infusions with the highest PA values, between 434 and 1171 µg/kg. The mean PA content was 159 µg/kg; the lowest was equal to the LOQ, 5 µg/kg. Retrorsine-N-oxide was the predominant analyte found, according to the results of other studies [40,41]. There are other studies stating that chamomile and lemon balm showed the highest PA amount, e.g., Schulz et al., with a maximum PA content of 53–1595 µg/kg [44]. In the scientific report by Mulder et al. [6], which collected the results of analyzed herbal infusions from different European counties, PAs were detected in all analyzed dried herbal infusions at concentrations between 1304 and 4894 µg/kg, higher in rooibos and peppermint. The general PA pattern observed in the EU samples (Sen-type and Lyc-type) suggests that species of *Senecio*, *Boraginaceae*, and *Heliotropium* may all be relevant contaminant species in this type of herbal infusion.

#### 2.2.5. Dried Herb Samples

Among 24 samples of dried herbs analyzed, 13 showed a PA content ≥ LOQ, of which 5 exceeded the maximum level (1000 µg/kg) [20]. Among them, the sample with the highest value of PAs was marjoram, 4678 µg/kg; three samples of oregano with values between 1043 and 4678 µg/kg; and one sample of dry borage, with a PA content of 2959 µg/kg. The mean total content was 1089 µg/kg, and the lowest was 5 µg/kg, equal to the LOQ. The oregano and marjoram belonged to the Lamiaceae family, and the dry borage to the Boraginaceae family, all PA-producing plants [3,45]. Lycopsamine-N-oxide was the most predominant analyte. In agreement with our findings, oregano was the sample with the highest PA levels in a large number of studies [46,47,48,49]. In the study by Izcara et al. (2020) [46], 56% of the oregano samples contained lyc-type compounds, but in contrast to our results, the major contribution was echimidine-N-oxide, while in a recent work by Izcara et al. (2022), PAs classified as heliotrine- and senecionine-type were the most prevalent in dried herbs analyzed [50]. In the Kaltner et al. [47] and Picron et al. [42] studies, heliotrine-type compounds were quantitatively predominant.

The occurrence of heliotrine-type alkaloids is usually associated with co-harvesting with *Heliotropium* spp. and *Borago* spp., while sen-type PAs are usually associated with species of the Asteraceae family, especially the *Senecio* family [42,47,50].

#### 2.2.6. Fresh Borage Leaf Samples

*Borago officinalis* (Figure 2) is the only PA-producing plant intentionally consumed [51]. Cultivated for its seed oil and rich in essential fatty acids, this Mediterranean herb is also used in salads, teas, and pasta [52,53].

All samples of fresh borage leaves were naturally contaminated with PAs. Only one sample exceeded the limit (750 µg/kg), with a value of 3410 µg/kg; this value corresponded to the concentration of lycopsamine-N-oxide, as shown in Appendix A, the predominant analyte in all samples analyzed. The mean PA content was 530.8 µg/kg, and the lowest was 5 µg/kg, equal to the LOQ.

Unfortunately, there are very few reports in the literature on detecting PAs in fresh borage leaves. Larson et al. [52,54] were the first to identify some PAs, such as lycopsamine, in fresh and dried borage leaves. Their investigation showed that in roots, PAs are mainly present as the free base, whereas fresh leaves mainly contain N-oxides, in line with the results of this study.

## 3. Materials and Methods

### 3.1. Sampling

A total of 602 commercial samples were collected on the Italian market between 2019 and 2022. Honey (*n* = 320) and bee pollen (*n* = 130) were collected from several Italian beekeepers before being put on the market. Fresh borage leaves (*n* = 13), dried herbs (*n* = 24), dried herbal infusions (*n* = 41), and dried tea (*n* = 74) came from all Italian regions. The analyzed samples are summarized in Table 3.

### 3.2. Materials and Reagents

Analytical standards of all 35 PAs and their N-oxide were provided from Phytolab (Vestenbergsgreuth, Germany): Echimidine, Echimidine-N-oxide, Echinatine, Echinatine-N-oxide, Europine, Europine-N-oxide, Heliosupine, Heliosupine-N-oxide, Heliotrine, Heliotrine-N-oxide, Indicine, Indicine-N-oxide, Integerrimine, Integerrimine-N-oxide, Intermedine, Intermedine-N-oxide, Lasiocarpine, Lasiocarpine-N-oxide, Lycopsamine, Lycopsamine-N-oxide, Retrorsine, Retrorsine-N-oxide, Rinderine, Rinderine-N-oxide, Senecionine, Senecionine-N-oxide, Seneciphylline, Seneciphylline-N-oxide, Senecivernine, Senecivernine-N-oxide, Senkirkine, Spartioidine, Spartioidine-N-oxide, Usaramine, and Usaramine-N-oxide. Of the 35 analytes, some are isomers and have been classified into five groups: the Sn group (Sn, Sv, and Ir), the Ly group (Ly, Im, Id, En, and Rn), the Sp group (Sp and St), the Em group (Em and Hs), and the Rt group (Rt and Us); the same applies to the N-oxides (Table 1) [20]. Methanol (LC-MS grade) was from VWR Chemicals (Rosny-sous-Bois-cedex, France); sulfuric acid (96%), ammonia (30%), and acetonitrile (LC-MS grade) were from Carlo Erba Reagents (Val de Reuil Cedex, France); formic acid was from Carlo Erba Reagents (Milano, Italia); and ammonium formate (analytical grade) was from Sigma-Aldrich (St. Louis, MO, USA). Ultrapure water was obtained from an EvoQua Water Technologies system (Diessechem, Milano). QuEChERS (Quick, Easy, Cheap, Effective, Rugged, and Safe) reagents, consisting of 4 g magnesium sulfate, 1 g sodium chloride, 1 g sodium citrate, and 0.5 g disodium hydrogen citrate sesquiydrate, were purchased from Agilent Technologies (Santa Clara, CA, USA).

### 3.3. Working Solutions

Stock standard solutions for each pyrrolizidine alkaloid and their N-oxide (1000 µg/mL) were prepared in methanol. Working standard solutions containing a mixture of PAs at different concentrations were prepared in water/methanol (95:5 *v*/*v*). All solutions were stored at −20 °C.

### 3.4. Sample Preparation

The samples were extracted with an acid solution and purified by SPE or QuEChERS, depending on the matrix. A 2.5 ± 0.1 g aliquot of homogenized honey was extracted with 15 mL of sulfuric acid 0.1 M. After vortex-mixing and 45 min of horizontal shaking, 15 mL of acetonitrile and QuEChERS extraction reagents were added. An aliquot of dried herbs, herbal infusions, tea (1 ± 0.1 g), and pollen (2 ± 0.1 g) was appropriately weighed and homogenized. For the bee pollen, wet homogenization was carried out. Each sample was extracted with 25 mL of sulfuric acid 0.05 M and, after vortex-mixing, shaken for 45 min and ultracentrifuged at 20,000 rpm at room temperature for 10 min. The extracts were loaded in SPE columns [21,55].

#### 3.4.1. SPE Procedure

A 10 mL aliquot of each sample (dried herbs, dried tea, and dried herbal infusions) was transferred and passed through Oasis MCX solid-phase extraction cartridges (Mixed-Mode Cation eXchange sorbent, 60 mg/3 mL, Waters) that were preconditioned with methanol (3 mL) and activated with sulfuric acid 0.05 M (3 mL). An aliquot of the extract was then loaded into the cartridge. The cartridges were then washed with 3 mL of water and 3 mL of methanol and eluted with 2.5 mL of methanol containing 2.5% ammonia. The eluates were evaporated to dryness with a gentle flow of nitrogen in a water bath at 40 °C, dissolved in 1 mL of water/methanol (95:5 *v*/*v*), and transferred into the vials for LC-MS/MS analysis. A quality control sample, i.e., spiked sample at LOQ—5 µg/kg—for pollen, dried tea, dried herbs, and dried herbal infusions, was assessed at every batch analysis [55].

#### 3.4.2. QuEChERS Procedure

Homogenized honey samples were extracted with QuEChERS reagents and shaken for 30 min. After 10 min of 3000 rpm of centrifuge at room temperature, an aliquot of the supernatant extract was dried with a gentle flow of nitrogen in a water bath at 40 °C. The dry extract was dissolved in 1 mL of water/methanol (95:5 *v*/*v*) and transferred into the vial for LC-MS/MS analysis. A quality control sample, i.e., spiked blank sample at LOQ—1 µg/kg—was assessed at every honey batch analysis.

### 3.5. LC-MS/MS System and Chromatographic Conditions for Analysis

The analysis was conducted on an Acquity ultra-performance liquid chromatograph (UPLC) coupled with a Quattro Premiere XE triple quadrupole mass spectrometer (Waters, Milford, MA, USA). The chromatographic separation was achieved on an Acquity UPLC C8 100 cm × 2.1 mm, 1.7 µm column (Water Corporation, Milford, MA, USA). Data acquisition and processing were carried out by TargetLynx software v. 4.2. The mobile phase consisted of 5 mM ammonium formate and 0.1% formic acid in water (A) and methanol (B) [55]. The mobile phase gradient consisted of B from 5% to 20% for 10 min, from 20% to 50% for 5 min, and returning to the initial condition in 0.5 min and holding for 1.5 min. The total run time was 17 min. The flow rate was 0.3 mL/min. The injection volume was set at 10 μL. The ESI source operated in positive ionization mode with the following instrumental parameters: capillary voltage of 1.0 kV, cone voltage of 40 V, source temperature of 120 °C, and a desolvation temperature of 450 °C. The conditions of ionization and PA fragmentation were identified by continuous infusion of the tuning solutions and gradual adjustment of the parameters.

### 3.6. Quantification

According to SANTE/12089/2016, the PAs were identified by the retention time, a target ion, and two qualifier ions. The retention time was within ±0.2 min of the reference peaks. The peaks showed similar peak shapes and overlapped with each other. The ion ratio was within ±30% of the average of the calibration standards from the same sequence. The peaks were within the linear range of the detector with an S/N ≥ 3 [56]. According to the EURL-MP guidance document on performance criteria for plant toxins, multi-level calibration curves—concentration levels from lowest to highest—were prepared for quantification of the PA content of unknown samples [23]. The least-squares regression method was used to extrapolate PA concentrations. Honey calibration curve concentration levels were 0.5-1-2.5-5-10-25 ng/mL. Dried tea, herbal infusions, and dried herbs calibration curve concentration levels were 0.5-1-2.5-5-10-25-50 ng/mL, while, for pollen, the solvent calibration curve was prepared at concentration levels of 0.5-1-2.5-5-10-25-50 ng/mL. The chromatograms of the working standard solutions with a mixture of PAs and PANOs (5 µg/kg) are shown in Figure 3 and Figure 4, respectively, where the phenomenon of coelution is evident for the following analytes: Em-Hs, Sp-St, Im-Id, Ly-Rn-En, Sn-Ir, RtNO-UsNO, ImNO-IdNO, and SpNO-StNO.

### 3.7. Performance Evaluation

The LC-MS/MS method was examined for specificity, recovery rates, linearity, repeatability, within-laboratory reproducibility, and LOQ, according to the EURL-MP guidance document on performance criteria for plant toxins and Regulation 401/2006/EC [23,57]. Specificity was checked, and the presence of interferences was verified by analyzing 20 blank samples for each matrix. The deviation in the back-calculated concentrations of calibration standards from the true concentrations using the calibration equation was not more than ±20%. The LOQ (1 µg/kg for honey and 5 µg/kg for pollen and dried products of tea, herbal infusions and herbs, and fresh borage leaves) was settled based on a signal-to-noise ratio S/N ≥ 5 (LOQ), and subsequently, the performance for recovery and precision was validated/verified. The LOD (0.5 µg/kg for honey) was settled based on a signal-to-noise ratio, S/N = 3 [23]. Repeatability and recovery were assessed by analyzing blank samples fortified with PAs at different concentrations in six replicates per level, 1-10-25 µg/kg (25-250-625 µg/kg for the sum) for honey, 5-500-1000 µg/kg (125-12,500-25,000 µg/kg for the sum) for pollen, and 5-400-800 µg/kg (3.75-25-125 µg/kg for the sum) for dried herbs, dried tea, dried herbal infusions, and fresh borage leaves (Appendix A). To determine within-laboratory reproducibility, the same experiment was performed in two additional sessions. According to SANTE/12089/2016, implemented by 01/01/2017 [56], the LC-MS/MS method’s selectivity was evaluated by acquiring the data in MRM mode and monitoring one precursor ion and two daughter ions for each molecule (Table 4). 

## 4. Conclusions

In this paper, a monitoring study on the occurrence of PAs in different commercially available food on the Italian market is presented. A total of 602 samples were collected and analyzed. All matrix types showed PA contamination ranging between 29 and 92%. The values found were compared with the new maximum levels established by EU Commission Regulation 2023/915 (25 April 2023). In particular, 14% of the contaminated samples exceeded the maximum levels. These samples were represented by dried herbs and dried tea (*Camellia sinensis*). For dried herbs, oregano featured the matrix with the highest values of PAs. These findings align with the EU RASFF notifications over the last few years. A low percentage of the samples were represented by fresh borage leaves, which are consumed as vegetables in different Italian regions. *Borago officinalis* is a PA-producing plant and showed the highest alkaloid levels (3410 µg/kg). Bee pollen and honey may be exposed to PAs, as confirmed by the analysis because these alkaloids are commonly concentrated in the flowering part of plants frequented by bees for nectar and pollen. The most predominant analytes in these food categories were also studied. They were classified as the lycopsamine type, occurring in the most widespread PA-producing plants in Italy. In conclusion, this monitoring showed that some food items may contain considerable amounts of PAs. The availability of data on PAs may be a starting point for assessing the contribution of these contaminants to one’s total exposure while considering vulnerable population groups. In addition to the food products studied in the present paper, the PA content in other food products, less studied or not regulated, such as animal products other than honey, should be monitored. Furthermore, it is important to analyze the possible effect of cooking techniques on PA contents. Continuous efforts should be made to minimize or prevent PAs’ occurrence in foodstuffs by applying good agricultural and harvest practices to ensure a high level of human health protection.

## Figures and Tables

**Figure 1 molecules-28-05346-f001:**
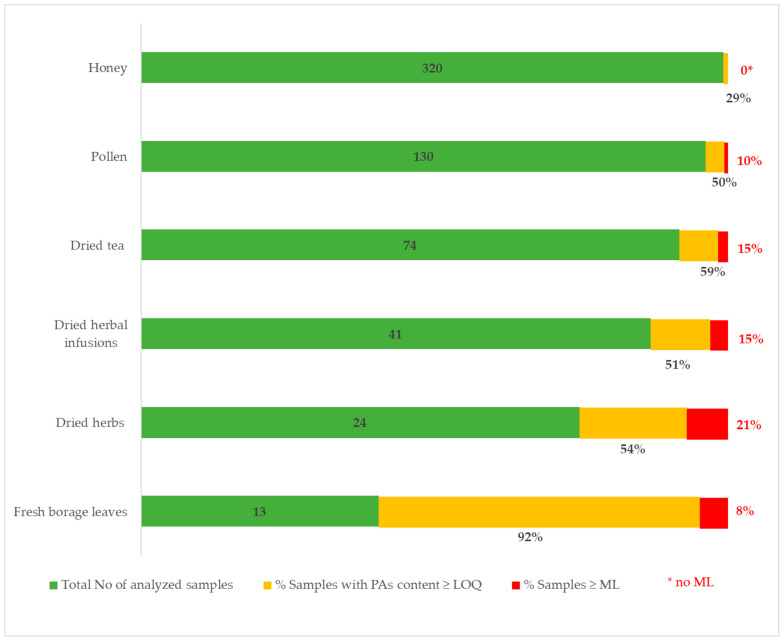
Total number of analyzed samples, percentage of samples with a PAs content ≥ LOQ, and percentage of samples exceeding maximum level (ML) for each matrix analyzed. No ML for honey in EU Regulation 2023/915.

**Figure 2 molecules-28-05346-f002:**
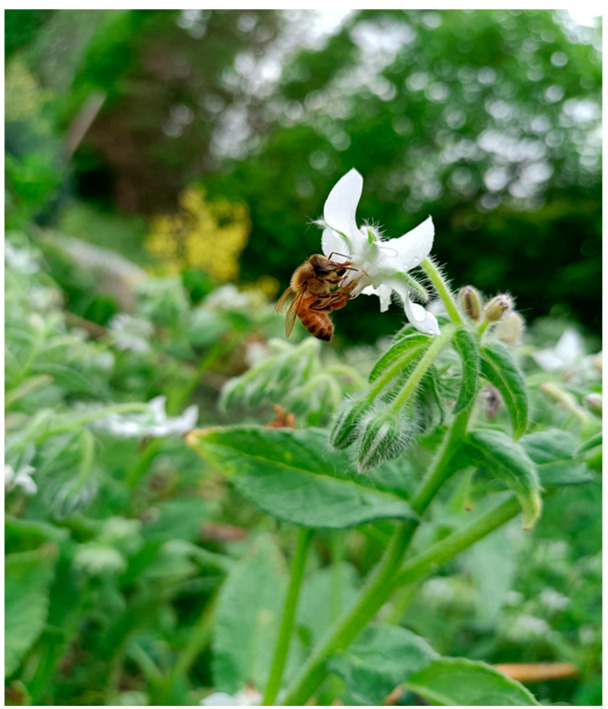
Bee on a flower of *Borago officinalis* (IZSLER, 2021).

**Figure 3 molecules-28-05346-f003:**
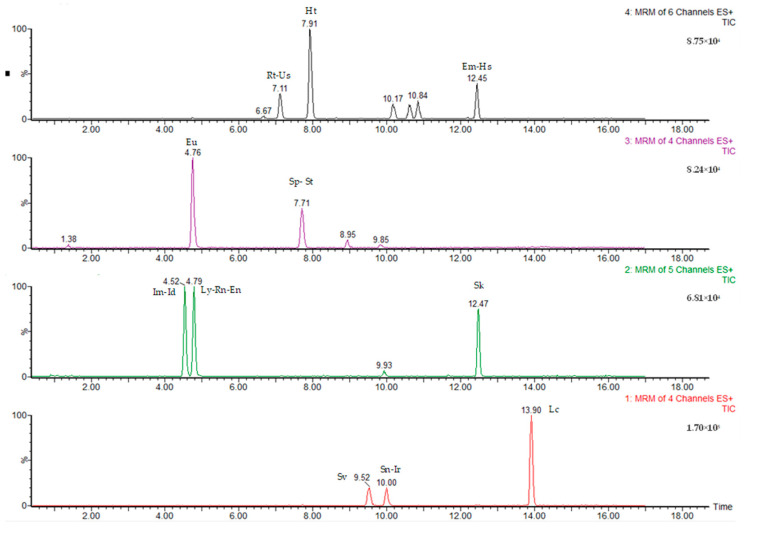
Chromatogram of a standard mixture of 18 PAs (5 ng/mL).

**Figure 4 molecules-28-05346-f004:**
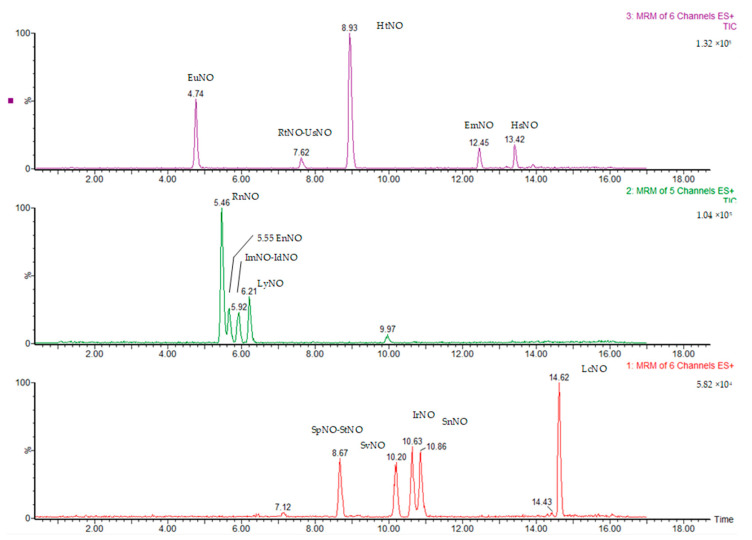
Chromatogram of a standard mixture of 17 PANOs (5 ng/mL).

**Table 1 molecules-28-05346-t001:** Classification of PA/PANO subtypes and groups investigated and their botanical origins.

PAs—Subtype	PAs—Group ^1^	Botanical Origins
Senecionine-type	Sn-group, Rt-group, Sp-group, and their N-oxides and Sk	*Senecio* spp. (Asteraceae family) *Crotalaria* spp. (Fabaceae family)
Lycopsamine-type	Em-group, Ly-group, and their N-oxides	*Echium* spp., *Borago officinalis* (Boraginaceae family) *Eupatorium* spp. (Asteraceae family)
Heliotrine-type	Eu, Ht, Lc, and their N-oxides	*Heliotropium* spp. (Boraginaceae family)
Monocrotaline-type	Mc and Mc-N-oxide (n.i. ^2^)	*Crotalaria* spp. (Fabaceae family)

^1^ Sn-group: senecionine (Sn), senecivernine (Sv), integerrimine (Ir); Rt-group: retrorsine (Rt), usaramine (Us); Sp-group: seneciphylline (Sp), spartoidine (St); senkirkine (Sk); Em-group: echimidine (Em), heliosupine (Hs); Ly-group: lycopsamine (Ly), intermedine (Im), indicine (Id), echinatine (En), rinderine (Rn); europine (Eu); heliotrine (Ht); lasiocarpine (Lc); monocrotaline (Mc); and their N-oxides: SnN-group: SnNO, SvNO, IrNO; RtN-group: RtNO, StNO; SpN-group: SpNO, StNO; EmN-group: EmNO, HsNO; LyN-group: LyNO, ImNO, IdNO, EnNO, RnNO; EuNO; HtNO; LcNO; McNO) [21]. ^2^ Not investigated; in Mediterranean countries, *Crotalaria* spp. is not found [22].

**Table 2 molecules-28-05346-t002:** The PA concentrations range of contaminated samples and predominant analytes found.

Matrix	LOQ ^1^ (µg/kg)	Sum PA/PANO Min Content (µg/kg)	Sum PA/PANO Max Content (µg/kg)	PAs Maximum Level ^2^ (µg/kg)	Predominant Analyte	Sub-Type
Bee pollen	5	6 ± 1.2	10,168 ± 2033	500	Echinatine-N-oxide	Lyc ^4^-type
Dried herbs (e.g., dried borage, marjoram, oregano)	5	11 ± 2.2	4678 ± 935	1000	Lycopsamine-N-oxide	Lyc-type
Fresh borage leaves	5	5 ± 1	3410 ± 682	750	Lycopsamine-N-oxide	Lyc-type
Dried tea (*Camellia* *sinensis*)	5	5 ± 1	1346 ± 269	150	Retrorsine-N-oxide	Sen ^5^-type
Herbal Infusions (e.g., lemon balm, chamomile, peppermint)	5	5 ± 1	1171 ± 234	400	Retrorsine-N-oxide	Sen-type
Honey	1	1 ± 0.2	121.1 ± 24	/ ^3^	Echimidine	Lyc-type

^1^ Limit of Quantification; ^2^ Commission Regulation (EU) 2023/915—the maximum levels refer to the PAs sum [19]; ^3^ No maximum level has been set for honey in the Commission Regulation (EU) 2023/915; ^4^ Lycopsamine-type; ^5^ Senecionine-type.

**Table 3 molecules-28-05346-t003:** Collection of analyzed samples.

Food Categories	Matrix	Botanical Origin	No. of Samples	%
Fresh borage leaves	Fresh borage leaves	*Borago officinalis*	13	2
Dried herbs	Rosemary, marjoram, basil, borage, oregano, coriander, and cinnamon	*Rosmarinus officinalis*, *Origanum majorana*, *Ocimum basilicum*, *Borago officinalis*, *Origanum* L., *Coriandrum sativum* L., and *Cinnamomun verum* L.	24	4
Dried herbal infusions	Lemon balm, chamomile, and herbal teas made from mallow, peppermint, karkadè, ginger, and fennel seeds	*Melissa officinalis*, *Matricaria chamomilla*, *Malva sylvestris* L., *Mentha x piperita*, *Hibiscus sabdariffa* L., *Zingiber officinalis*, and *Foeniculum vulgare* Mill.	41	7
Dried tea	Black and green tea	*Camellia sinensis*	74	12
Pollen	Bee pollen	Unknown ^1^	130	22
Honey	Monofloral (acacia, chestnut, linden, rapeseed, honeydew, betony, and sunflower) and multi-floral	*Robinia pseudoacacia*, *Castanea sativa*, *Tilia* L., *Brassica napus*, *Sulla coronaria*, *Stachys officinalis* and *Heliantus annuus* L.	320	53

^1^ The species of pollen samples analyzed was unknown; no melissopalynological analysis was performed.

**Table 4 molecules-28-05346-t004:** LC-MS/MS parameters for all PAs/PANOs (CE: collision energy, Q: quantifier ion, q: qualifier ion).

Pyrrolizidine Alkaloids	MH^+^	CE	*m*/*z*	Q, q
Sn group	336.2	25	120.2	Q
		25	138.0	q
Ly group	299.7	20	138.0	Q
		25	156.0	q
Ht	314.1	20	138.0	Q
		25	156.0	q
Eu	330	20	138.0	Q
		15	156.0	q
Sk	366.1	30	122.0	Q
		25	167.9	q
LyN group	316.1	25	172.0	Q
		25	138.0	q
HtN	330.2	25	172.0	Q
		25	111.0	q
EuN	346.2	25	172.0	Q
		20	328.1	q
Lc	412.1	25	120.1	Q
		18	220.0	q
Sp group	334	25	120.1	Q
		25	138.0	q
Em group	398.6	20	119.9	Q
		15	220.4	q
Rt group	352.1	25	120.0	Q
		25	138.3	q
SnN group	352.1	25	94.0	Q
		30	118.0	q
LcN	428.1	30	254.0	Q
		25	94.0	q
SpN group	350.1	30	94.0	Q
		25	120.0	q
EmN group	414.2	30	254.0	Q
		25	220.0	q
RtN group	368.3	30	94.0	Q
		20	120.0	q

## Data Availability

Not applicable.

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
