# Peer review of "Pyrrolizidine Alkaloids in Food on the Italian Market"

_molecules, 2023, doi:10.3390/molecules28145346_

Round 1

Reviewer 1 Report

The manuscript entitled Pyrrolizidine alkaloids in Food on the Italian market aimed to carry out a survey about the monitorization of these alkaloids in different available food products on the Italian market between 2019-2022. Although the occurrence of PAs is currently a hot research topic, the manuscript lacks originality, the data presented is scarce, and regulation issues are not updated. For these reasons this reviewer has to reject the manuscript. 

Main concerns about the manuscript:

1) There is a new EU regulation 2023/915 from April that has updated EU 1881/2006 and 2020/2040. Some limits have changed, the authors should revised this new regulation and change the reference 2020/2040 for 2023/195.

2) The authors state in their manuscript that they analyzed and quantified 35 analytes (18 PAs and 17 PANOs). However, as showed in the chromatograms provided, they have coelution of some isomers, for this reason, they mention they use different groups, so they are quantifiying some compounds together. Therefore, they have not the complete separation of the 35 analytes, and this should be indicated and specified in the manuscript. Accordingly, Lines 88-89 are not correct.

3) SANTE/12089/2016 has a more recent regulation of 2019, this should be updated for the validation part.

4) The authors indicated the criteria specified in the validation guidelines (e.g. ion ratio within +/- 30%, recoveries between 70-120%, precision >20%, etc...) however data about these parameters is not provided anywhere in the manuscript, not even as a supplementary material.

5) How was the s/n calculated to estimated the LOQ and LOD? Some examples of chromatograms should be at least provided to see how it has been determined. Likewise, why the authors established the LOQ as s/n > 5? Usually is 10 times higher than the s/n, and 3 times for LOD. 

6) which sample matrices were used for the validation? Authors do not explained the type os samples used to carry out the evaluation of the different analytical parameters. 

7) No information about the samples (e.g. types of honey, types of teas, etc...) are provided by the authors. 

8) Authors do not explain how the recovery was calculated. They refer to blank samples fortified, but how it was then the recovery calculated. This should be especified. 

9) the abstrat is to general, it should be revised according to the work aim. 

10) there is no individual data of the analytes in the different samples, not even in supplementary material.

11) Discussion of the results is very poor. With a huge number of samples, at least some statistical stufy is required.

12) There is no information of mena values, or even medians of the different samples analyzed. 

13) Line 142: the highest value found was 121.1 ug/kg, authors did not specify the honey sample, and then they talk about multifloral (8 ug/kg) and monofloral (45.3 ug/kg). This should be revised.

14) Figure 1 and 2 should showw some sd or error bars

15) Table 3 should show a +/- of the sum content

16) Table 2 should shoe a sd value in the mean recovery

17) Figure 3 is irrelevant

18) Figure 4 is irrelevant if no mass spectrum is provided. This chromatograms should be provided with a fortified sample spiked at the same concentration level, and see if the ion pattern keeps the ion relation. Same for figure 6

19) Table 4 is irrelevant. This information is provided in the text in section 3.1. Nonetheless, the table could be complete with the information of the different types of samples analyzed (the different types of honey, their origin, the same for the teas etc.). 

20) Why author provide for the linear regression the units in ng/mL, if they are constantly refering the LOQ, data values etc. in ug/kg. This should be revised.

21) Authors indicated than borage is contaminated, while it is a PA-producing plant. Therefore, it can not be contaminated. The presence of PAs in borage is natural and it is obvious that they found these alkaloids in the borage leaves. Actually, it is  surprising that authors state that 12 out of 13 borage samples were contaminated. It is not possible that one of the samples did not show PAs, as it is a natural secondary metabolite of this family plant. I hardly recommend the authors to revise this, and also to consider to ommit that borage samples are contaminate, because they contain PAs in a natural way. 

22) in the abstract author mention contamination through soil, but is not the only pathway, as explained in the introduction. This should be revised in the abstract.

23) The introduction is very similar to the information provided in reviews already published about the occurrence of PAs. I suggest to center the introduction in the aim of the study, and obviously update it with the correct regulation.

Minor comments:

1) english language should be revised

2) abbreviations should be revised, because some times they are mention after they have already been mentioned in the text. For instance, abbreviations of analytes are included in section 3.2, while some of these analytes are being mentioned in section 2. Also for LOQ

3) The aim of the work is usually included in the introduction, not in the conclusion section

English language should be carefully revised, there are some errors, e.g. line 244 " a total of 602 commercial samples was collected" it shoul be were collected.

Reviewer 2 Report

The English expression is clear and unambiguous. It can effectively convey the background, methods, results, and conclusion of the study.

Reviewer 3 Report

The content of this paper was to report the investigation results of the Pyrrolizidine alkaloids (PAs) in food on the Italian market. It provided very useful information about the (PAs) in food. Some problems need to be considered before it could be accepted as a research paper.

1. The analysis was conducted on an Acquity ultra-performance liquid chromatograph coupled to Quattro Premiere XE triple quadrupole mass spectrometer. Please illustrate its measurement uncertainly. About the LOQ and LOD, how to ensure these values? These values were obtained from the literature or practical calibration procedure?

2. What are the sources of Pas in the food? Please supply some information about this food safety problem.

3. How about the PAs values in other countries, especially in the EU?

4. Please provide more information about the Pas from different sources. How to sample them and how to improve this Phenomenon? In this paper, the authors only tell the investigation results. In this stage, the content of this paper is liked a laboratory report, not a research paper.  Please enhance the content, especially in the section of Discussion and Conclusion.

Minor editing of English language required

Reviewer 4 Report

 - This work makes us ask: how to differentiate between scientific research and routine scientific tests to examine the safety and quality of foods and medicines according to official regulations approved by countries.

-        -   Material and methods: Why is this method not used as part of the control procedures for medicine and food by the official authorities before the food products reach the market?

-          -  “PAs occur in different parts of the plants especially in the seeds and inflorescences, and their content depends on a large number of factors (species, plant organ, harvest and climatic conditions)". What is the most important factor? Insect herbivores?

-          - What is the differences between these toxins and phytoalexins. Or , these toxic alkaloids not belong to phytoalexins?.

-          - In table 1, it showed the wild plants origins that contains pyrrolizidine alkaloids (PAs) as main constituents. Why these alkaloids found in the other plant families such as these tested products and how they found in animal products, this must be improved in the discussion.

-          It is better to focus on the natural origin itself (e.g. chamomile), and not on the form of the product (infusion). I suggest, in addition to commercial name and mode preparation of each product, it must added the scientific names of species and family of the origin for each tested plant and animal product in the tables. The determination of the scientific name is the first important step for such analytical examination.

-          Table 2. Add word “honey” to word “pollen” It is also better to add classification of tested samples according to their the main natural sources: animal and plant sources.

Minor editing of English language required

Round 2

Reviewer 1 Report

I appreciate the effort made by the authors to improve the manuscript considering the reviewers comments. However, I still considering the article has no enough quality and originality to be pubished in the journal. 

Authors: We are talking about samples analyzed between 2019-2022 so we referred to the Regulation that came into force last year 2020/2040. In the new Regulation 20223/915, however, the limits have not changed. However, we think it is correct to specify that that regulation has been repealed and that the new regulation is now in force, so thanks you for the advice

It does not matter when the samples were collected. Currently there is a new regulation and the article is intended to be published after the publication of this regulation. Therefore, the article needs to be adapted to the current situation, nonetheless the results provided are soundless and not updated. 

Authors: Regulation 2020/2040, like 2023/915, states that there are certain isomers that coelute, and therefore have the same molecular weight and transitions. According the Regulation, it is not possible to separate all the isomers, and the legal limits are given by the sum of PAs/PANOs. However, we have pointed this out in lines 380-38

It is true there are isomers, and in this work the authors do not have the complete separation of the 35 analytes. COnsequently, it is not possible to individually detect and quantify them. Therefore, it is not correct that they have validated the method for the 35 analytes. Some results correspond to a sum of compounds because of the coelution. So Authors can not say their method achieve the separation and analysis of the 35 compounds, because some of them are being identify together. This is of utmost importance and has to be clear in the manuscript.

Authors: The SANTE Regulation to which we refer is the Guidance Document on the identification of mycotoxins in food and feed, SANTE/12089/2016 Implemented by 01/01/2017, not 2019.

What I am referring to is that the SANTE document of 2017 is not updated to the current situation. Therefore, authors should update their data to the new criteria of the update SANTE document, As indicate previously with the regulation, if authors are not following the latest updates of the validation guidelines or of the regulation, their data is not reliably supported and it is meanless for the scientific community.

Authors: According to EURL-MP-guidance doc_003 (version 1.1) Guidance document on performance criteria (draft 17th September, 2021), for official control purposes determination of the limit of detection (LOD) is not required. For the determination of the LOQ we relied on S/N approaches: LOQ = concentration corresponding to S/N ≈ 5-10. It was calculated with 6 replicates of fortified samples, was settled on the basis of a signal-to-noise ratio S/N ≥ 5 (LOQ), and subsequently the performance for recovery and precision was validated/verified, as added in lines 395-398.

Is this document reference in the validaton part? why the authors indicate the SANTE but not this document. The explanation given should be included with its reference. 

Authors: According to EURL-MP-guidance doc_003 (version 1.1) Guidance document on performance criteria (draft 17th September, 2021), the recovery was calculated with: Rec (%) = x/xref ×100, where: x = measured concentration (for spiked samples corrected for background concentration if applicable) xref = reference concentration (the concentration of spiked sample). We specified in the text that we refer to the EURL document in lines 390-39

Why the authors refer again to this document, when they indicated they are using the criteria of the SANTE? I believe things are not being explained correctly

Authors: The authors consider the figures to be an enrichment to the text.

Figures enrich the text when they show some relevant and interesting, and information is useful, otherwise they just fill the gap

Authors: abbreviations should be revised, because some times they are mention after they have already been mentioned in the text. For instance, abbreviations of analytes are included in section 3.2, while some of these analytes are being mentioned in section 2. Also for LOQ

According to the manuscript preparation guidelines, Abbreviation are always defined at first mention, not when the authors considered is is appropiate to defined them. 

Revise the language

Reviewer 2 Report

I am writing to inform you that the author has made the requested revisions to the article according to my requirements. After reviewing the revised version, I recommend accepting it for publication.

Reviewer 3 Report

All problems have been revised adequately.

 Minor editing of the English language required

Reviewer 4 Report

There is previous studies similar to this study about “Pyrrolizidine alkaloids in food and feed on the Belgian market”. This has not been added or discussed. In a previous study about Europe markets, samples were taken from different European countries, including Italy. What is the differences between these studies results?

1)      Huybrechts, B. and Callebaut, A., 2015. Pyrrolizidine alkaloids in food and feed on the Belgian market. Food Additives & Contaminants: Part A, 32(11), pp.1939-1951.

minor
